Kinome state is predictive of cell viability in pancreatic cancer tumor and cancer-associated fibroblast cell lines

Berginski Matthew E. 1
Jenner Madison R. 1 2
Joisa Chinmaya U. 3
Herrera Loeza Gabriela 2
Golitz Brian T. 4
Lipner Matthew B. 1 2
Leary Jack R. 2 5
Rashid Naim 2 6
Johnson Gary L. 1 2
Yeh Jen Jen jen_jen_yeh@med.unc.edu 1 2 7
Gomez Shawn M. smgomez@unc.edu 1 3
1 Department of Pharmacology, University of North Carolina at Chapel Hill , Chapel Hill , United States of America
2 Lineberger Comprehensive Cancer Center, University of North Carolina at Chapel Hill , Chapel Hill , NC , United States of America
3 Joint Department of Biomedical Engineering at the University of North Carolina at Chapel Hill and North Carolina State University , Chapel Hill , United States of America
4 Eshelman Institute for Innovation, University of North Carolina at Chapel Hill , Chapel Hill , NC , United States of America
5 Department of Biostatistics, University of Florida , Gainsville , FL , United States of America
6 Department of Biostatistics, University of North Carolina at Chapel Hill , Chapel Hill , NC , United States of America
7 Department of Surgery, University of North Carolina at Chapel Hill , Chapel Hill , NC , United States of America
Karakülah Gökhan
Electronic publication date: 2024 Aug 28
Publication date: 2024
Volume: 12
Electronic Location ID: e17797
Received 2023 Apr 21; Accepted 2024 Jul 2
Copyright: ©2024 Berginski et al.
Copyright year: 2024
Copyright holder: Berginski et al.
License: This is an open access article distributed under the terms of the Creative Commons Attribution License, which permits unrestricted use, distribution, reproduction and adaptation in any medium and for any purpose provided that it is properly attributed. For attribution, the original author(s), title, publication source (PeerJ) and either DOI or URL of the article must be cited.
License URL: https://creativecommons.org/licenses/by/4.0/

Keywords: Cancer, Cell signaling, Machine learning, Predictive modeling, Drug response, Kinase inhibitor treatment, Tumor microenvironment, Drug sensitivity, Pancreatic cancer

Funding: National Institutes of Health T32CA071341 (MRJ) R01CA193650 (JJY) R01CA199064 (JJY) U24DK116204 (GLJ, SMG) U01CA238475 (GLJ, SMG) R01DK109559 (SMG) R01CA233811 (SMG) This work was supported by the National Institutes of Health—T32CA071341 (MRJ), R01CA193650 (JJY), R01CA199064 (JJY) U24DK116204 (GLJ, SMG), U01CA238475 (GLJ, SMG), R01DK109559 (SMG), R01CA233811 (SMG). The funders had no role in study design, data collection and analysis, decision to publish, or preparation of the manuscript.

==============================
Numerous aspects of cellular signaling are regulated by the kinome—the network of over 500 protein kinases that guides and modulates information transfer throughout the cell. The key role played by both individual kinases and assemblies of kinases organized into functional subnetworks leads to kinome dysregulation driving many diseases, particularly cancer. In the case of pancreatic ductal adenocarcinoma (PDAC), a variety of kinases and associated signaling pathways have been identified for their key role in the establishment of disease as well as its progression. However, the identification of additional relevant therapeutic targets has been slow and is further confounded by interactions between the tumor and the surrounding tumor microenvironment. In this work, we attempt to link the state of the human kinome, or kinotype, with cell viability in treated, patient-derived PDAC tumor and cancer-associated fibroblast cell lines. We applied classification models to independent kinome perturbation and kinase inhibitor cell screen data, and found that the inferred kinotype of a cell has a significant and predictive relationship with cell viability. We further find that models are able to identify a set of kinases whose behavior in response to perturbation drive the majority of viability responses in these cell lines, including the understudied kinases CSNK2A1/3, CAMKK2, and PIP4K2C. We next utilized these models to predict the response of new, clinical kinase inhibitors that were not present in the initial dataset for model devlopment and conducted a validation screen that confirmed the accuracy of the models. These results suggest that characterizing the perturbed state of the human protein kinome provides significant opportunity for better understanding of signaling behavior and downstream cell phenotypes, as well as providing insight into the broader design of potential therapeutic strategies for PDAC.

Introduction

While improvements in outcome for pancreatic ductal adenocarcinoma (PDAC) have occurred in the last decade with 5-year survival increasing from 5% to 12%, more therapeutic options are clearly needed (American Cancer Society, 2023). Major barriers to developing effective therapies have been the low tumor cellularity in PDAC and the uniquely hostile, poorly vascularized and highly fibrotic tumor microenvironment. Two tumor-intrinsic subtypes of PDAC have been identified, classical and basal-like, with the basal-like subtype being consistently associated with poorer survival and less responsive to first-line cytotoxic combination therapies including FOLFIRINOX or gemcitabine plus nab-paclitaxel (Rashid et al., 2020; Aung et al., 2018; Moffitt et al., 2015; O’Kane et al., 2020). Distinct from tumor subtypes, two molecular subtypes of PDAC stroma have also been identified: “activated” which is associated with poor outcome as well as “normal” stroma. We have previously shown that cancer associated fibroblasts (CAFs) are a contributory cell type within the stroma compartment of PDAC and that CAFs may significantly alter response to therapy by impeding drug diffusion or disrupting microenvironment homeostasis (Moffitt et al., 2015; Provenzano et al., 2012; Olive et al., 2009; Özdemir et al., 2014; Toste et al., 2016). These results point to the need for new therapeutic approaches that, in addition to providing better activity against the tumor, also provide enhanced efficacy against the appropriate tumor microenvironment.

Since the introduction of imatinib (Gleevec), kinase inhibitors have emerged as a focus for targeted therapy development, in part due to the numerous roles played by kinases in cellular signaling as well as the connection between their dysfunction and disease (Druker et al., 2001). A number of kinase signaling pathways have been identified as components of PDAC initiation and progression, including the hallmark RAS/RAF/MAPK, AKT/PI3K, as well as aberrant signaling from various growth factor receptors (TGFβ, EGFR, VEGFR) (Orth et al., 2019; Murthy, Attri & Singh, 2018). Linked with their role in disease, the druggability of kinases has led to strong growth in the development of kinase inhibitors, with over 72 achieving FDA approval (Roskosk Jr, 2023).

In parallel with inhibitor development has been the growth of assay techniques capable of quantively assessing kinase response to these drugs. In particular, recent proteomics techniques including multiplexed inhibitor beads linked with mass spectroscopy (MIB/MS) and Kinobeads enable the ability to assess the state of the protein kinome en masse (Duncan et al., 2012; Collins et al., 2018; Klaeger et al., 2017). This approach utilizes broad-spectrum type 1 kinase inhibitors covalently linked to Sepharose beads to pull down and enrich for kinases. In a control or untreated setting, kinases present in the sample are free to bind Kinobeads. However, in the presence of a kinase inhibitor, competitive binding between the inhibitor and Kinobead will take place where the strongest interaction wins and Kinobead-bound kinases are depleted. The bound kinases are then digested with trypsin, identified using liquid chromatography tandem mass spectrometry (LC-MS/MS), and quantified using the MS intensity (Fig. 1A). Quantification of the dynamic response of the perturbed kinome provides a novel platform to profile drug targets and design new therapies.

Figure 1 Model overview and distribution of drug response data.

(A) Visual abstract of the generated predictive drug response model of PDAC. (B) Distribution of cell viability values for each of the three cell lines included in this study. (C) Distribution of cell viability across the inhibitors overlapping between the screening data and the Klaeger et al. (2017) compounds.

As an initial attempt to assess the predictive capability of kinome behavior and potentially expand the availability of novel therapeutic options in PDAC, here we describe a modeling effort that links the measured state of the human kinome, or “kinotype”, with the downstream phenotype of cell response in treated PDAC tumor and CAF cell lines. More specifically, we utilized our internal collection of kinase inhibitor response data measured at six doses on three patient-derived lines, with one cell line representing a CAF and the other two representing the primary tumor. Inhibitors from this screen that synergize with the combination therapy, FOLFOX were published previously Lipner et al. (2020), whereas the focus of this work is on single-agent kinase inhibitors. These PDAC-specific viability response data were then linked with a unique public data set that assessed the broad proteomic response of the human kinome to kinase inhibitor treatment using Kinobeads (Klaeger et al., 2017). As a result, we were able to link cell response in the tumor and CAF cell lines to the inhibition levels of 332 kinases in response to 59 kinase inhibitors applied at five different doses (Fig. 1A).

We found that prediction of the cellular response to a given inhibitor was achievable through classification models, with receiver operating characteristic (ROC) scores of ∼0.7 for all cell lines using a random forest approach. We further used these models to identify specific kinases whose behavior in response to perturbation drove the majority of response outcomes in these cell lines, pointing to the possibility of identifying tumor- and CAF-specific target candidates for further investigation. Finally, we experimentally validated these models by treating cells with previously untested drugs and comparing responses to model predictions. We found that linking kinome inhibition states with cell viability provides a more precise representation of drug response since it captures kinome perturbations versus baseline expression. Since the latter does not always indicate activity, especially in the case of kinases, these proteomic approaches more closely represent drug activity allowing for the discovery of better drug targets. Importantly, kinome states are interpretable and could help determine markers of response or stratify patient cohorts by those most likely to benefit from therapy based on associated kinome responses. Together, this systems view of kinome behavior and its linkage with downstream phenotypes suggests potential opportunities for the identification of novel drug targets and targeted therapy options for PDAC.

Methods

Portions of this text were previously published as part of a preprint (https://doi.org/10.1101/2021.07.21.451515).

Cell lines

Pancreatic cancer cell lines were derived from patient-derived xenografts as described in methods published previously (Moffitt et al., 2015). Briefly, pancreatic tumor samples from deidentified patients were obtained under protocols approved by the UNC IRB. Tumors were subcutaneously implanted into the flanks of 6-8 week old female NSG mice. Mice were passaged according to protocols approved by the UNC Institutional Animal Care and Use Committee. At the time of passage, a section of the tumor was cut into approximately 3-mm pieces and rinsed with PBS containing penicillin and streptomycin. The tissue was minced with a gentleMACS Dissociator (Miltenyi Biotec) and incubated for 30 min in Collagenase/Dispase (11097113001; Roche) solution or Human Tumor Dissociation kit (Miltenyi Biotec) for 1 h.

CAF cell lines were derived from pancreatic tumor samples from deidentified patients obtained under protocols approved by the UNC IRB. Tumors were dissociated using the protocol described above. Dissociated tissue was resuspended in DMEM/F12 with 10% FBS.

CAFs were immortalized for continual culturing by infecting with hTERT (pBABE-hTERT-puro) within 2-weeks of CAF line establishment. HEK293T cells were transfected with pCL-10A1 to generate retrovirus using X-tremeGENE 9 DNA transfection reagent (Sigma). Growth media was replenished after twenty-four hours and viral supernatants were harvested 48 h post-transfection. In a T25 flask, CAF cells were stably transfected with 4 ml of retrovirus and 4 ug/ml Polybrene (Sigma). Immortilized CAFs were grown out and selected for using puromycin-containing media.

All cell lines were maintained in DMEMF12 (Gibco) supplemented with 10% FBS. All cell lines were cultured in an incubator at 37 °C with 5% CO2 and were regularly tested for mycoplasma contamination and cell line identity by short-tandem repeat testing.

PDAC kinase inhibitor screen

P0422-T1, P0411-T1, and P0119-T1 CAF cells were seeded in 384-well flat-bottom plates (Greiner) at densities of 2000, 1800, and 450 cells/well, respectively. Twenty-four hours after seeding, the 176 epigenetic and kinase inhibitor library (EpiKin176, published previously Bevill et al. (2019)) was stamped across six doses: 10 µM, 3 µM, 1 µM, 300 nM, 100 nM, 10 nM using the Biomek FXP Laboratory Automation Liquid Handling Workstation (Beckman Coulter). DMSO was used as the vehicle control at a concentration of 0.1% on cells. Four biological replicates were conducted for each cell line. Synergistic effects of EpiKin176 with the combination therapy FOLFOX were assessed previously Lipner et al. (2020). Seventy-two hours post-treatment, cells were lysed with CellTiter-Glo (Promega) per the manufacturer’s protocol. Luminescence was read using the PHERAstar FS microplate reader (BMG Labtech). Data were normalized to the DMSO-only control to calculate relative viability.

Validation studies of predicted Klaeger drugs

P0422-T1, P0411-T1, and P0119-T1 CAF cells were seeded at 3500 cells/well in white flat-bottom 96-well plates (Corning). Twenty-four hours after seeding, cells were treated with masitinib, ripasudil, AT-13148, RGB-286638, PHA-793887, lestaurtinib, AT-9283, KW-2449, K-252a, PF-03814735, and XL-228. Each drug was dosed at the same eight concentrations used in the Klaeger study: 30 µM, 3 µM, 1 µM, 300 nM, 100 nM, 30 nM, 10 nM and 3 nM. Seventy-two hours post-treatment, cells were lysed with CellTiter-Glo (Promega) per the manufacturer’s protocol. Luminescence was read using the PHERAstar FS microplate reader (BMG Labtech). Three technical replicates (triplicate wells per plate) and 2 biological replicates were collected for each cell line and drug pair. Quality checks were performed to look at the data distribution and the presence of spatial bias on a plate. One of the replicate runs of PHA-793887 and AT-9283 failed to meet this criteria and were removed from analyses. Data were normalized to the DMSO-only (0.1% on cells) control samples on each each plate to calculate relative viability.

General modeling methods

All of the models developed in this study were produced using the R programming language. The tidymodels modelling framework was used for both the regression and binary classification models (Kuhn & Wickham, 2020). During model development we used a custom cross validation approach, which left one compound out of model training and used the remaining data for model development. For each model type tested, we also conducted a hyperparameter search over 100 combinations with latin hypercube sampling (Iman, Helton & Campbell, 1981) across all the investigated hyperparameters. We used the ROCR package to collect ROC and precision-recall curve (PRC) values and curves (Sing et al., 2005). The functions available in the tidyverse package were extensively used to organize the data sets (Wickham et al., 2019).

Cell viability classification modeling

We tried three types of classification models: support vector machine (SVM) (Karatzoglou et al., 2004), random forest (Wright & Ziegler, 2017), and XGBoost (Chen & Guestrin, 2016). We sampled over the same set of hyperparameters for the random forest and XGBoost models as in the regression models. For the SVM model, we sampled over the following hyperparameters and ranges:

• Cost of predicting prediction error: [−10 to 5] with log2 transformation

• Polynomial degree: [1–3]

• Kernal scaling factor: [−10 to −1] with log10 transformation

Software and data availability

We have made all scripts and processing code for this project available through github under the BSD license: https://github.com/gomezlab/PDACperturbations and Zenodo: https://doi.org/10.5281/zenodo.11623371. This repository also contains the experimental data used to build our models.

Results

Data organization and distribution of viability values

In order to build a collection of models tailored to predicting cell response in our PDAC cell lines, we first needed to organize and combine the data sets. This was divided into two arms, the first dedicated to collecting and organizing the data from our internal drug screen and the second to organizing the kinase inhibition data provided by Klaeger et al. (2017).

As the cell viability screen used a larger set of compounds than the Klaeger et al. (2017) collection for testing, the screening data was filtered to only include overlapping compounds. The Klaeger et al. (2017) collection was processed in turn, starting with the kinase inhibition table available in the Supplemental Materials (Table S1, Kinobeads subsheet). This spreadsheet is organized to show the kinases and a few non-kinase genes whose inhibition states are affected by each of the kinase inhibitors tested by Klaeger et al. (2017). As such, a wide range of kinase targets are included across the range of compounds tested. For the purposes of organizing the data to enable downstream machine learning applications, we assumed that all kinases not listed with a given compound were unaffected by this compound. The baseline value for an unaffected kinase is a ratio of one, so all missing compound/kinase combinations (95% of combinations) were filled in with one. We also found a small set of compound/kinase combinations where a single concentration was missing, so we filled these values in with the average value of the two closest concentrations included in the assay. Finally, we also found a few values with unexpectedly high ratio values (up to around 25), so we truncated all values in the collection to the 99.99% percentile (ratio value of 3.43).

With our two critical data sets collected and organized, we combined these data sets by matching the inhibitors and the corresponding concentrations. The screening data was collected at six inhibitor concentrations (1e−8, 1e−7, 3e−7, 1e−6, 3e−6 and 3e−5 all in M) and all of these concentrations were matched in the Klaeger data except for 1e−5 M, which we removed from the combined data. After this filtering and matching process, we had data matches across 59 compounds. Since the full Klaeger compound set consists of 229 compounds having effects on 520 kinases, we also determined how many of these kinases were affected by our screening compound matches. This step indicated that 188 of the kinases showed zero variance in our screening compound set, so we removed these kinases from our modeling efforts. This left 332 kinases with some amount of variance in the screening compound matched data set.

The overall distribution of cell viabilities across each cell line was similar, with the majority of the compounds having little effect on cell viability (Fig. 1B). This was mirrored in the distribution of cell viabilities across the compounds, with most of the compounds having a mean viability of above 90% (Fig. 1C). However, several of the compounds do demonstrate greater variation in the viability affects, with alvocidib, CUDC-101, and AT-7519 showing the greatest variability in viability across all concentrations. The drug screen data, now matched with corresponding kinase inhibition states as measured in the Klaeger data, allowed us to build a set of models to predict the cell viability using the kinase inhibition state results. We next explored using classification models to link changes in the human kinome state, as induced through kinase inhibitor treatment, with downstream cell viability as described below.

Cell viability binary classification

We built a set of classification models centered around predicting cell viability from the kinase inhibition values. To convert the cell viability prediction problem into a binary prediction problem, we thresholded the viability values at 90% cell viability. This thresholding divided the cell the viability values into two classes with 45.5% of the CAF line data below 90% viability, 35.7% under the cutoff in the P0422-T1 line and 40.2% under the cutoff in the P0411-T1 line. We selected three model types, random forests, SVM, and XGBoost for testing. To find the optimal model for each cell line and model type, we used a cross validation approach combined with hyperparameter scanning. The cross validation approach we used was based around leaving the data pertaining to a single compound out of the training data and then building a model with the data from the remaining compounds. The primary reason we used this cross validation approach was to support our goal of building a model capable of predicting the cell viability for compounds which were not included in the model’s training data.

Given that each of these models have a set of hyperparameters that could effect the results of the cross validation testing, we also conducted a hyperparameter search over 100 different hyperparameter combinations for each of the three model types tested. From this collection of cross-validation predictions covering a range of possible hyperparameter sets and model types, we selected the best model, hyperparameter set and cell line on the basis of the ROC score (Fig. 2A). Of note, we also developed regression models using generalized linear model (GLMnet), random forest, and XGBoost to predict cell viability of the treated cell lines, however, none of the regression model types were able to accurately predict the cell viability of a left out compound (RMSE ranged from 13.9 to 19.1) (Fig. S1).

Figure 2 Classification models show promise in predicting drug response in PDAC tumor and CAF cell lines.

(A) ROC curves for each of the best performing hyperparameter sets for each model type with the inset text indicating the area under the curve for each model type. (B) Precision–recall curves for each model type with the inset text indicating the area under the curve. The cell line used to develop each model set is indicated by the gray titles above each plot. Dashed lines represent performance for random guessing.

As seen in Fig. 2, each modeling method performed similarly in terms of ROC and precision–recall with the random forest model performing the best across all models. The best single model was the P0119-T1 CAF random forest achieving an ROC score of 0.711 compared to ∼0.71 and ∼0.67 for P0411-T1 and P0422-T1, respectively. A similar pattern in performance was also observed in the precision–recall curves. This consistency of performance gave us confidence that the random forest method was most likely to yield accurate predictions when used to make predictions on compounds where we had no corresponding cell viability results. Thus, we opted to use random forest and the corresponding sets of optimal hyperparameters for each cell line to predict binarized cell viability probabilities for the inhibitors examined in the Klaeger set, but which had not yet been tested in our cell lines.

Known kinase targets of PDAC are recapitulated

With the random forest method and associated hyperparameters selected, we next built models for each of the cell lines using the full set of compound and concentration matched data. To examine kinases within these models whose behavior were considered the most important for the resulting random forest models, we gathered variable importance data (Greenwell & Boehmke, 2020) for each cell line (Figs. 3A–3C). Note that as referenced earlier, non-kinases can be captured via Kinobeads and therefore, may also be identified as important in the models; e.g., EPHB6 (EPH receptor B6), EIF3J (eukaroytic translation initiation factor 3 subnunit J), and AZI2(NF- κB activating kinase-associated protein 1). These plots revealed that the top 30 proteins considered important for each model was not a constant list, with only nine kinases in common across all feature lists (Fig. 3D), suggesting there are different kinase vulnerabilities for each of the cell models. Among the shared kinase features of all three cell lines are known PDAC targets including MAPK(MAP2K1/2, MAP4K5) and AURKB (Figs. 3A, 3B and 3C). Interestingly, EGFR, also a well studied target in PDAC, is a top feature for both the tumor lines but not the CAF line (Figs. 3A and 3B).

Figure 3 Variable importance results for optimal random forest models.

A set of barplots showing the 30 protein targets with the highest ranked importance for each optimized random forest model and associated importance measure in the P0411-T1 (A), P0422-T1 (B) and P0119-T1 CAF (C) models. (D) Upset plot indicating the overlaps in the top 30 proteins in the three models.

MAPK

The MAPK pathway regulates proliferation, differentiation, and gene expression. MAPK is downstream of KRAS- a constitutively activated protein mutated in over 90% of PDAC cases (Waters & Der, 2018). When MAPK signaling is deregulated or overactive, the tumor continues growing without restraint. For these reasons, MAPK is a well-known target in PDAC and other cancers. Several kinase inhibitors have been tested in clinical trials for PDAC that directly target MAPK (NCT04892017, NCT05907304, NCT05585320, NCT05630989, NCT04005690, NCT01155453), upstream regulators of MAPK (NCT01077986, NCT04985604) or downstream effectors (NCT04566393, NCT02608229, NCT04386057, NCT05039177). These trials do not include pan-RAS or RAS mutant inhibitors such as the KRAS G12C inhibitors sotorasib or adagrasib which recently achieved FDA approval for non-small cell lung cancer. Taken together, these data provide support in the validity of the model by identifying kinase features that are specific to PDAC.

AURK

Aurora kinase (AURK) A/B are serine-threonine kinases involved in angiogenesis, epithelial-mesenchymal transition, metastasis, and cell cycle progression by regulating mitotic entry (Wan et al., 2008; Hong et al., 2022; Liu et al., 2016; Nigg, 2001). Overexpression and deregulation of AURKA/B is prevalent in many cancers and in some cases is associated with disease outcome. In PDAC, high AURKA expression (greater than median AURKA expression), is predictive of significantly lower survival in The Cancer Genome Atlas (TCGA) Research Network: https://www.cancer.gov/tcga. Due to its involvement in uncontrolled proliferation and negative correlation with survival, AURKA has emerged as a promising target with inhibitors advancing through clinical trials. Combining alisertib (AURKA inhibitor) with chemotherapy is rationalized to have increased efficacy due to more specific inhibition of mitosis which could decrease off-target toxicities seen in chemotherapies. In clinical trials for PDAC and other solid malignancies, alisertib has been tested with first-line therapies, gemcitabine or nab-paclitaxel in which several patients achieved stable disease status by RECIST criteria including one PDAC patient having a partial response (NCT01924260, NCT01677559). Alisertib is expected to be investigated further in future clinical phase studies.

EGFR

Epidermal growth factor receptor (EGFR) is a transmembrane growth factor receptor in the tyrosine kinase family. External ligand binding activates EGFR which leads to receptor dimerization and initiation of several signaling cascades including as RAS/RAF/MAPK and AKT/PI3K that ultimately stimulate growth. EGFR ranked in the top five kinases for the PDAC lines but was absent in the CAF model, making it a potential tumor-specific target. Overexpression of EGFR is found in upwards of 90% of PDAC tumors and is associated with more advanced disease (Ueda et al., 2004; Tobita et al., 2003). The EGFR inhibitor, erlotinib, remains the only approved targeted therapy for PDAC. Together, these results suggest that the models are identifying known, clinically-relevant kinases in PDAC that warrant further exploration as drug targets.

Novel, understudied targets of PDAC are identified

Besides finding well-known PDAC targets, several poorly understood kinases were identified and highly ranked in the feature selection process. Such understudied kinases have been identified by the US National Institutes of Health as high-potential therapeutic targets as part of their Illuminating the Druggable Genome Project (IDG) (Oprea et al., 2018; Gomez et al., 2024). Specifically, CSNK2A1/3 was identified as highly ranking in the P0422-T1 tumor line (Fig. 3B) and the kinases CAMKK2 and PIP4K2C were identified in the P0119-T1 CAF line (Fig. 3C).

CSNK2A1/3

CSNK2A1/3 (Casein Kinase 2 Alpha 1/3) are involved in WNT, Hedgehog (Jiang, 2017; Purzner et al., 2018), and NFKB signaling as well as cell cycle progression and apoptosis (Trembley et al., 2009). Multiple studies in pancreatic cancer cells have shown that targeting CSNK2, either by pharmacological inhibition or gene silencing, leads to increased apoptosis and decreased NFKB transcription activity (Kreutzer, Ruzzene & Guerra, 2010; Giroux et al., 2009). Furthermore, silencing either CSNK2A1 or CSNK2A2 in PDAC cells sensitized cells to gemcitabine, one of the first line therapies for PDAC (Kreutzer, Ruzzene & Guerra, 2010). This growth inhibitory phenotype translated to the in vivo setting where PDAC xenograft models with CSNK2A1 silenced by siRNA had decreased tumor volume compared to siRN control mice. The tumor volume was further decreased when co-silenced with the WNT activator, PAK7, or MAP3K7. Of note, silencing CSNK2A1 significantly increased apoptosis in MiaPaCa2 cells but not in the other 11 non-pancreatic human tumor cell lines (Giroux et al., 2009). This shows a potential tumor selective vulnerability that can be exploited in PDAC or a subset of PDAC patients since CSNK2A1/3 and CSNK2B were top features in only one of the tumor cell models, P0422-T1 (Fig. 3B). Furthermore, the diversity of kinases identified as important within the models shows that the relationship of a given kinase with cell viability varies even in closely related cell lines.

CAMKK2

Calcium/calmodulin-dependent protein kinase kinase 2 (CAMKK2) is a serine/threonine intracellular receptor kinase. CAMKK2 activity is stimulated by Ca2+ which leads to phosphorylation of CAMK effectors (CAMK1/2) as well as AMPK (AMP-activated protein). Together with its effectors, CAMKK2 regulates processes related to the cytoskeleton, inflammation, and glucose homeostasis (Mukherjee et al., 2023). Interestingly, CAMKK2 is a top feature found only in the P0119-T1 CAF and not the tumor models (Fig. 3C). Recently, CAMKK2 signaling was identified as a driver of CAF macropinocytosis in response to depleted glutamine and cytosolic Ca2+ in PDAC. Proteins macropinocytosed by CAFs were not only preserved intracellularly to sustain CAF growth but were extracellularly secreted which directly promoted tumor cell proliferation (Zhang et al., 2021). Therefore, CAMKK2 represents a CAF-specific kinase vulnerability which may have therapeutic relevance in inhibiting tumor-CAF crosstalk.

PIP4K2C

Phosphatidylinositol-5-phosphate 4-kinase type 2 gamma (PIP4K2C) is an understudied, lipid kinase with its function remaining relatively unknown compared to the PIP4K2A and PIP4K2B isoforms. PIP4K2A and PIP4K2B serve as second messengers to signaling pathways and are typically found in the cytosol and nucleus, respectively (Clarke, Wang & Irvine, 2010; Ciruela et al., 2000). While PIP4K2C localization is still unclear, there has been a report of PIP4K2C being in the cis-Golgi, suggesting PIP4K2C may be involved in cellular trafficking (Clarke, Emson & Irvine, 2008). Since the PIP4K2 isoforms have different cellular localizations, it is likely they regulate distinct cellular processes.

Other reports suggest PIP4K2C may play a role in immune function as it is a substrate of mTOR1 (Mackey et al., 2014). Knocking out PIP4K2C (Pip4k2c−/−) in mice elicited a reactive immune environment with higher proportions of helper T cells and less regulatory T cells (Shim et al., 2016). Thus, inhibiting PIP4K2C may be a strategy to mount an immune response. As a top feature specific to the P0119-T1 CAF model (Fig. 3C), perhaps PIP4K2C is a regulator of the tumor microenvironment due to its involvement with the immune system and cellular trafficking. However, more research is needed to characterize the function and regulation of PIP4K2C before it is deemed a putative target.

Taken together, these results suggest that the developed models can identify cell-specific and potentially novel, understudied drug targets to investigate in the context of PDAC. Furthermore, the diversity of highly-ranked kinases identified in feature selection within the models suggests that the relationship of a given kinase with cell viability varies greatly even in closely related cell lines.

Using random forest models to predict drug effects on cell viability

With the models built for predicting the cell viability with each of our cell lines, we then used these models to predict cell viability for the clinical kinase inhibitors in the Klaeger et al. (2017) collection which were were previously unseen and therefore not used to build the original models. There were 165 inhibitors at eight concentrations in this set (1320 total combinations). Each cell-line specific model was used to predict the likelihood that a given compound and concentration combination would cause cell viability to go below 90% (Fig. 4A). Overall, the P0119-T1 CAF probability predictions of decreasing viability below 90% for the untested inhibitors were the highest with an average value of 0.448, followed by the P0411-T1 line at 0.409, and the P0422-T1 line at 0.371.

Figure 4 Random forest model predictions for clinical kinase inhibitors.

(A) Histograms showing the distributions of the probability of each compound and concentration combination having a below 90% cell viability effect. The distributions are split by cell line with the cell line name above each plot. (B–E) Plots showing the top three compound probabilities of a below 90% cell viability effect optimized for (B) predicted no drug effect (lowest average probabilities), (C) predicted drug effect (highest average probabilities), (D) differential drug response across concentrations(highest range of probabilities) and (E) differential cell line response (largest probability differences between cell lines). In parts B–E, plots are labeled by each predicted compound and colored by cell line shown in the legend below this collection of plots.

From this collection of predictions, we extracted four categories that we thought would be especially relevant for future validation and testing efforts (Figs. 4B–4E). These categories of compound predictions are: no predicted drug effect (lowest average probability of viability being below 90%, Fig. 4B), drug effect (high probability of viability being below 90%, Fig. 4C), differential drug response (highest range of probabilities across concentrations, Fig. 4D), and differential cell line response (largest probability differences across cell lines, Fig. 4E). The predictions across cell lines were similar within the predicted (no drug effect Fig. 4B) and predicted drug effect groups (Fig. 4C), whereas there was a larger range of cell line probabilities in the differential drug response across concentrations (Fig. 4D) and cell lines (Fig. 4E), indicating that there are differences in both drug activity and cell line response. These categories of compounds show that the remainder of the Klaeger et al. (2017) compound set is predicted to have varying cell viability effects.

Most notably, several of the compounds identified as having differences across concentrations (Fig. 4D) show large differences in the predicted effects at the mid-dose range with increasing likelihood of viability effects as the compound concentration increases. This predicted effect, that mirrors a standard dose response curve, is due entirely to modifications in the kinase inhibition states as none of the models use compound concentration as an input value. These predictions highlight how modeling methods can be utilized in drug development to filter out negative inhibitors or identify inhibitors with preferential activity against tumor versus CAFs, for example, depending on the project aims. With the model predictions collected, we selected the top two to three inhibitors within each of the predicted groups in Figs. 4B–4E for follow up drug response studies to test our predictions.

Validating a subset of model predictions

To validate the prediction models, we selected 11 previously untested compounds within our prediction groups Figs. 4B–4E and performed a small scale cell viability screen with the tumor and CAF cell lines. We conducted the validation screen at all eight of the Klaeger compound concentrations (n = 3 technical replicates, n = 2 biological replicates) yielding 264 cell line, compound, and concentration combinations (Fig. 5). We assessed these results using several different methods. First, we compared the cell viability values collected in the validation assay to the probabilities produced by each of the models and found a clear trend, with the viability values decreasing as the model probabilities increased (Fig. 5A). Since the models are all predicting the probability that a given experiment will produce cell viabilities below 90%, the curve trend was in the expected downward-right direction. We also thresholded the cell viability values at 90% cell viability (as done with the training data) and produced ROC and PRC curves for the validation results (Fig. 5B). The ROC and PRC curves each demonstrate that the overall predictions for the validation compounds are performing as expected compared to the the cross validation results (Fig. 5B). These global methods for quantifying model performance confirmed that the majority of the predictions made by the models were accurately observed in the followup validation screen.

Figure 5 Validation results for previously untested kinase inhibitors.

(A) Comparison between the measured cell viability values and the predicted probability of that experiment yielding a cell viability value below 90%. The blue line shows a loess fit through the data. (B) ROC and PRC curves for the validation data set thresholded at 90% viability. (C) ROC and PRC curves for each of the validation compounds tested (n = 3 technical replicates, n = 2 biological replicates). Dashed lines represent performance for random guessing.

We also subdivided the validation results by the tested compounds and produced individual compound ROC and PRC curve results (Fig. 5C). Again, models were quite accurate for eight out of 11 inhibitors tested. The models performing the worst were those predicted to have little effect on cell growth (masitinib, ripasudil and AT-13148). In the case of the compounds predicted to have a small effect, the largest effect on cell growth was observed at the highest tested compound concentration (30 µM), which may be causing off target effects. Because off target effects may not be fully captured in the mass spectroscopy based assay technique used by Klaeger et al. (2017) the models would be expected to be unable to correctly predict the decrease in cell viability in these cases. With this caveat concerning interpreting the predictions at the highest concentrations in mind, we conclude that the models were largely successful at predicting the cell viability effects of a novel collection of kinase inhibitors.

Discussion

In this study, we developed a collection of models that predict cell viability from kinase inhibition states and used them to predict the effect of inhibitor treatment on PDAC tumor and CAF cell line models. The kinase inhibition data was collected through a mass spectroscopy-based method which provided an unprecedented view of how kinase inhibitors effect the entire human kinome (Klaeger et al., 2017). This data has a wide range of potential uses and through this study we have connected this data to the results of a drug screening assay in PDAC cell lines. This drug screening effort overlapped with only a subset of the compounds and concentrations in the Klaeger et al. (2017) data set, but this amount of overlap was sufficient to build a collection of models which were capable of predicting the cell viability effects of kinase inhibitors. By examining the importance of each of the kinase inhibition states as determined by the models, we also found that a diverse set of well-studied and understudied kinase inhibition states were important for the modeling predictions. Of note, the dose of any compound is not used directly in the model. Rather, the generated models only use the kinase inhibition state data to perform inferences, with dose being indirectly encoded through the drug’s effect on kinase inhibition states at the specified dose. Using these predictions, we selected 11 additional compounds for validation screening and found that the model predictions were confirmed, with only a few exceptions. Overall, the linkage of kinome states and cell response enables the discovery of new kinase targets and provides broader insight into the cellular differences in kinase features or vulnerabilities which have applications in precision medicine. We have made all of the source code and data associated with this work publicly available through GitHub (https://github.com/gomezlab/PDACperturbations).

While this work focuses on a small number of cell lines, the methods developed here are independent of the cell lines studied. Further work could expand the cancer types covered by gathering preliminary data using a small set of kinase inhibitors to bootstrap a set of models corresponding to any cancer type of interest. Expansion to different types of cancer would also help to clarify the role of specific kinases or collections of kinases in cell viability.

This study was also limited to assessing the role of kinase inhibitors in cell viability, but cell viability is only one of a number of possible outcome measures that could be analyzed. Any high throughput assay that can be conducted in the presence of kinase inhibitors, such as measurement of metabolite concentrations or cellular imaging assays, could be adapted to use the framework described here to attempt to generate predictive models.

Finally, since the only input to the models developed in this study is kinase inhibition state, it should be possible to computationally combine the inhibition state vectors and then make inferences about the likely cell viability results of these novel compound combinations. This would in effect be a virtual synergy screen, which could cover a much broader range of compound combinations than would be experimentally feasible. In addition, such an approach could enable the prediction of drug combinations that would preferentially effect the tumor and tumor-promoting microenvironment.

These results suggest that there is significant information encoded in the protein kinome and point to the potential to further improve predictive capabilities through the inclusion of gene expression and related data. Furthermore, this systems view of the kinome and its integration into predictive models presents opportunities for the identification of new drug targets and the design of therapies in PDAC as well as other cancers.

Supplemental Information

File S1 Cell viability regression modeling methods

Figure S1 Regression models were ineffective at predicting cell viability

The results of the RMSE optimized regression models shown as hex binned heatmaps. Each column shows the optimized results for each cell line, while the rows show the type of model. The dot-dash lines show where a perfect set of predictions would appear, while the green lines show the linear best fit through the presented data points. The RMSE value is also presented in the lower corner of each plot.

Table S1 Protein target list from Klaeger et al., 2017

We would like to thank UNC research computing for access to computational resources.

Additional Information and Declarations

Competing Interests

Author Contributions

Data Availability

Shawn M. Gomez is an Academic Editor for PeerJ.

Matthew E. Berginski conceived and designed the experiments, analyzed the data, prepared figures and/or tables, authored or reviewed drafts of the article, and approved the final draft.

Madison R. Jenner conceived and designed the experiments, performed the experiments, analyzed the data, prepared figures and/or tables, authored or reviewed drafts of the article, and approved the final draft.

Chinmaya U. Joisa conceived and designed the experiments, performed the experiments, analyzed the data, prepared figures and/or tables, authored or reviewed drafts of the article, and approved the final draft.

Gabriela Herrera Loeza performed the experiments, authored or reviewed drafts of the article, and approved the final draft.

Brian T. Golitz performed the experiments, authored or reviewed drafts of the article, and approved the final draft.

Matthew B. Lipner performed the experiments, analyzed the data, authored or reviewed drafts of the article, and approved the final draft.

Jack R. Leary analyzed the data, authored or reviewed drafts of the article, and approved the final draft.

Naim Rashid conceived and designed the experiments, analyzed the data, authored or reviewed drafts of the article, and approved the final draft.

Gary L. Johnson conceived and designed the experiments, authored or reviewed drafts of the article, and approved the final draft.

Jen Jen Yeh conceived and designed the experiments, analyzed the data, authored or reviewed drafts of the article, and approved the final draft.

Shawn M. Gomez conceived and designed the experiments, analyzed the data, authored or reviewed drafts of the article, and approved the final draft.

The following information was supplied regarding data availability:

All scripts and processing code for this project are available at GitHub under the BSD license and Zenodo:

- https://github.com/gomezlab/PDACperturbations.

- Gomez S, Madison Jenner, Berginski M. 2024. PDAC Perturbations. Zenodo. https://doi.org/10.5281/zenodo.11623371. This repository also contains the experimental data used to build our models.

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
