# Peer review of "Kinome state is predictive of cell viability in pancreatic cancer tumor and cancer-associated fibroblast cell lines"

_PeerJ, doi:10.7717/peerj.17797_

## Round 0.1 · original submission · Major Revisions

I encourage the authors to carefully consider the reviewers' comments and revise your manuscript accordingly. The reviewers particularly highlighted the following issues:

Reviewer #1 recommended using the term 'human kinome' at least once in the manuscript, moving the negative results of the Cell Viability Regression section to supplementary material, and moving the section on novel candidate kinases to the discussion.

Reviewer #2 expressed concern about the lack of information provided about the kinome analysis approach used in the study. They suggested introducing the approach early in the manuscript or using a schematic diagram to make it easier to understand. Additionally, they recommended providing more information on the application and benefit of linking kinome states and drug response.

Reviewer #3 found the study to be a meaningful attempt to predict cell viability using the kinome inhibition state. However, they recommended several improvements, such as providing more evidence of the association between important kinase characteristics and cells and analyzing the reasons for showing certain figures.

·

Basic reporting

The term 'kinome' was used several times throughout the paper. Given that this a human study, it might be useful to use the term 'human kinome' at least once.
Typo at lines 146 and 147

Experimental design

No comment

Validity of the findings

The study is based on the assumption that the human kinome modulates information transfer, and its disregulation is critical for cancer. The effect of the kinome for PDAC viability was modelled via random forest and other classification methods. Most significant kinases for the model prediction were selected. Potential effects of kinase targeting drugs were predicted and experimentally validated in pancreatic cancer cell lines. Potential novel kinase targets were evaluated. Conclusion were clearly stated. Results of this study could be used to provide novel candidate drug therapies for pancreas cancer.

Additional comments

The details and the negative results of the Cell Viability Regression section of the Results could be left out to supplementary material. (lines 220-242)
The section for novel candidate kinases CAMKK2, POP4K2C and CSNK2A2/3 in Results, could be moved to discussion. (lines 294-324)

Reviewer 2 ·

Basic reporting

Overall, I found it is hard to understand the procedure in the current version of manuscript. I think it might be better to introduce and briefly provide information about approaches of kinome analysis work by Klaeger et al 2017 early in the manuscript (maybe introduction or first section of the result part). For me, I needed to read the reference article to figure out this work. Or a schematic diagram of work flow similar to those are shown in Klaeger et al 2017 might help make it easier to understand.

Minor comment: can the authors specify the comparative terms such as "significantly better" in line 252? And "the most importance" kinases in line 265, in what aspect, what is the behavior used?

Experimental design

This work aims to develop computational model that can predict viability of cancer cells toward drugs based on kinome state using kinome states dataset from previously published work to link with drug sensitivity data. This idea is interesting. However, I have concern that these two dataset were generated from different cell types with distinct genetic and kinomic background that involve with response to kinase inhibitors. How do the authors think about this point?

Since the authors use 3 types of cell including cancer cell lines and stromal cell lines and mentioned briefly about their biological relationship. It might be better to discuss more based on the authors' finding.

Can the authors state more about application and benefit of linking kinome states and drug response?

Validity of the findings

- Can the authors provide further explanation to why didn't the regression model work?
- How many biological and technical replicates of drug sensitivity assay?

·

Basic reporting

This manuscript reports a modeling attempt based on the kinome inhibition state to predict cell viability treated by compounds. Classification models for cell viability on three types of cells were successfully established, providing important kinase features for model prediction. In addition, the established optimal models was used to predict untested drugs and experimental validation was conducted. Overall, this is a meaningful attempt to predicting cell viability with a unique perspective by using the kinome inhibition state to characterize molecules.

Experimental design

However, there are still several aspects of this manuscript that could be improved.
1. The cell viability screen data source is ambiguous. Please clearly indicate the source of the data.
2. The poor performance of regression models is confusing, especially given that classification models can distinguish between differences in prediction probabilities caused by different drugs and concentrations. Is it possible to conduct further analysis, provide possible causes, and attempt to improve performance? Otherwise, the existence of the section “Cell Viability Regression” may be redundant.
3. Can you provide more evidence of the association between important kinase characteristics and cells in the “Identification of kinases driving model behavior” section? For example, some important kinase features may be potential therapeutic targets for the current cell model, and additional experimental validation or literature support should be listed. It is not enough to simply introduce the cancer-related reports of important kinases.
4. In the section “Using Random Forest Models to Predict Drug Effects on Cell Viability”, there is no analysis of Figure 5B, 5C, and 5E. Please explain the reasons for showing these figures and analyze them.
5. In the section “Validating a Subset of Model Predictions”, it is unclear how to select the 10 compounds. Please provide the reason for the selection.

Validity of the findings

Can you provide more evidence of the association between important kinase characteristics and cells in the “Identification of kinases driving model behavior” section? For example, some important kinase features may be potential therapeutic targets for the current cell model, and additional experimental validation or literature support should be listed. It is not enough to simply introduce the cancer-related reports of important kinases.

---

## Round 0.2 · accepted · Accept

Both reviewers of the revised manuscript have found the changes made to be sufficient for publication. However, Reviewer #2 has a minor suggestion regarding word choice, which the authors can address during the proofreading stage. I congratulate the authors for their work.

·

Basic reporting

no comment

Experimental design

no comment

Validity of the findings

no comment

Additional comments

All of my previous minor comments were addressed in the revised manuscript.

Reviewer 2 ·

Basic reporting

The authors have response to previous comments and the change is sufficient.

Experimental design

This work aims to develop computational model that can predict viability of cancer cells toward drugs based on kinome state using kinome states dataset from previously published work to link with drug sensitivity data. This idea is interesting. The authors had clarified the concern on discrepancy of cell types and mentioned about about relationship of cell types used and benefit of their finding.

Validity of the findings

The authors had responsed to previous comments:
- Can the authors provide further explanation to why didn't the regression model work?
Regression model performance was clearly stated in supplematry section.
- How many biological and technical replicates of drug sensitivity assay?
Number of replications stated in the authors response is sufficient.

Additional comments

I have minor suggestion for word choice but I will let this be the author judgement.

From previous comment: can the authors specify the comparative terms such as "significantly better" in line 252? And "the most importance" kinases in line 265, in what aspect, what is the behavior used? >>>>> I think using the word “the highest ranked kinases based on (these variables)” is more suitable as it avoids subjective words and is clear by itself of what it is. The authors might briefly introduce this group of kinases that they are the highest ranked based on what aspects. Then, using the word highest ranked kinases later on will be easy to understand.